# Long-term results of hypofractionation with concomitant boost in patients with early breast cancer: A prospective study

**Kitwadee Saksornchai**[1]*, **Thitiporn Jaruthien**[1,2], **Chonnipa Nantavithya**[1,2], **Kanjana Shotelersuk**[1], **Prayuth Rojpornpradit**[2]

1 Division of Therapeutic Radiology and Oncology, Faculty of Medicine, Chulalongkorn University, Bangkok, Thailand, 2 Division of Therapeutic Radiology and Oncology, Department of Radiology, King Chulalongkorn Memorial Hospital, Thai Red Cross Society, Bangkok, Thailand

* kitwadee.s@chula.ac.th

## Abstract

### Aim

To report the long-term local control and survival of patients with early breast cancer who had hypofractionated whole breast irradiation with concomitant boost (Hypo-CB).

### Methods and materials

Between October 2009 and June 2010, 73 patients with early breast cancer (T1-3N0-1M0) who underwent breast conserving surgery were enrolled into the study. Thirty-six of these participants received 50 Gy of conventional irradiation in 25 fractions over 5 weeks to the whole breast with a sequential boost to the tumor bed with 10–16 Gy in 5–8 fractions (Conv-SEQ). The other 37 participants received a hypofractionated dose of 43.2 Gy in 16 fractions with an additional daily concomitant boost (CB) of 0.6 Gy over 3 weeks (Hypo-CB).

### Results

At a median follow-up time of 123 months, ipsilateral local recurrence (ILR) was found in 3 participants, 1 of whom was in the hypofractionated group. All 3 ILR were true local recurrence (TR). There were no significant differences in the 10-year disease free survival (DFS) and 10-year overall survival rates (OS) between the conventional and hypofractionated groups (93.9% *vs.* 94.4%, p = 0.96 and 91.9% *vs.* 91.6%, p = 0.792, respectively).

### Conclusion

This study showed that the effectiveness, DFS and OS were comparable between hypofractionated whole breast irradiation with a CB and the conventional irradiation with a sequential boost.

**Data Availability Statement:** To protect potentially identifiable information on serious crimes, ethical approval is needed to access data. Data are available from ethics committee at faculty of

medicine, Chulalongkorn university (contact via medchulairb@chula.ac.th or kitwadee.s@chula.ac.th) for researchers who meet the criteria for access to confidential data.

**Funding:** The authors received no specific funding for this work.

**Competing interests:** The authors have declared that no competing interests exist.

## Introduction

After breast conserving surgery, followed by an adjuvant whole breast radiation treatment is the standard of care for early breast cancer. A number of phase III randomized studies have confirmed the benefits of adjuvant radiation in terms of reduction in the local recurrence and improved breast cancer specific survival [1–6]. The Early Breast Cancer Trialists' Collaborative Group meta-analysis revealed that adjuvant radiation reduced the 10-year risk of locoregional and distant recurrences by approximately 15%, while it reduced the 15-year risk of breast cancer death by approximately 4% [7]. The conventional radiation to the whole breast is 45–50 Gy, administered in 1.8–2 Gy per fraction doses over 5–6 weeks [1, 8].

Four pivotal randomized studies reported no significant differences in local control, toxicity and cosmesis between hypofractionated and conventional fractionation, hence, hypofractionation is now the most common treatment for patients with early breast cancer [9–15].

The effect of a tumor bed boost has been shown to improve local control rates. Bartelink et al. reported an absolute 4% lower 20-year cumulative incidence of ipsilateral breast cancer recurrence when a tumor bed boost is administered, but the incidence of skin fibrosis was higher in the tumor bed boost treated group [16, 17]. In another study, Lyon et al. showed consistent results that treatment with a tumor bed boost provided ipsilateral breast tumor control without much impact on toxicity [18].

Conventionally, a sequential tumor bed boost would be added to the whole breast irradiation which prolonged the overall treatment duration. In the START A trial, a 13 fraction regimen was used for 5 weeks, while the START B trial used a 15 fraction regimen for 3 weeks. Approximately 60% of the patients in START A and 40% from START B trials received a tumor bed boost, delivered in 5 sequential conventional fractions of 10 Gy. Both studies reported no increase in adverse events from tumor bed boost.

Simultaneous or concomitant boost for breast cancer has been adopted by many cancer institutes because the treatment duration is shorter and a higher dose is administered to the tumor bed which is the most frequent site of recurrence [19–21]. The toxicity and the cosmetic outcome are both acceptable. This technique administers whole breast irradiation and boost concomitantly. Recently, Krug et al. has published the data affirming that the toxicity is low and treatment compliance is high [22].

However, there is limited data on the long-term effects of hypofractionated CB treatment. At most, there is data up to 5 years post treatment. Thus, in this study, we report the 10-year follow-up of the efficacy of hypofractionated CB treatment.

## Materials and methods

### Study population

Between 2009 and 2010, a total of 73 patients with operable invasive breast cancer (T1-3N0-1M0) who had undergone breast conserving surgery were enrolled into the study. Thirty-six of these participants received conventional radiation (Conv-SEQ) and 37 participants received hypofractionated radiation (Hypo-CB). Both radiotherapy fractionation schemes are standard of care in our hospital. The treatment schedule was assigned by the radiation oncologists. Patients with T1-3N0-1M0, aged 18 or older and had resected tumor margins wider than 1 mm were enrolled into the study. We incorporated a boost to the tumor bed in this study (0.6 Gy for 16 fractions), by delivering a biologically effective dose (BED) of 96.36 Gy (with an $\alpha/\beta$ value of 4 Gy) which is equivalent to the sequential regimen of BED 90 Gy. The adjuvant systemic treatment was administered according to the NCCN Clinical Practice Guidelines in Oncology [23]. Patients with prior radiation to the thoracic area, patients with bilateral breast

cancer, patients with breast carcinoma with skin involvement, patients with connective tissue disease and patients requiring concurrent chemoradiation were excluded from the study. The study protocol was approved by the Research Ethics Committee of King Chulalongkorn Memorial Hospital and written informed consent was obtained from all patients. Patients who declined to take part in the study received the same standard of care as the patients who agreed to take part in the study.

## Definitions

Ipsilateral local recurrence (ILR) was defined as an in situ/invasive carcinoma that recurred after breast conserving therapy in the ipsilateral breast with or without regional recurrence or distant metastases. True local recurrence (TR) was defined as an invasive carcinoma that occurred in the same area/quadrant of the prior tumor bed and required tissue biopsy for pathological confirmation. Disease free survival (DFS) was defined as the time duration after radiotherapy during which no disease had been found. Overall survival rate (OS) was defined as the time after breast conserving therapy to death from any cause.

## Radiotherapy

The participants either received 50 Gy in 25 fractions followed by a sequential 10–16 Gy boost to the tumor bed or 43.2 Gy in 16 fractions with a concurrent boost of 0.6 Gy for each fraction. Radiation was administered over 5 and 3 weeks for the conventional and hypofractionated groups, respectively. The details of the radiation technique have been previously described [24]. Briefly, all of the participants were treated in the supine position on a breast board with arms above the head. Whole breast irradiation was performed either using two-dimensional (2D) radiation technique with two 6–10 MV tangential fields, with or without wedges, or three-dimensional (3D) conformal technique. An en-face electron beam was used for tumor bed boost, prescribed at 90% isodose line. Eclipse TPS version 7.3.10 was used to calculate the distribution of the radiation dose. The VARIAN Linear Accelerators machines were used to treat all of the participants (The Varian 21EX, 23EX, RapidArc, Clinac iX or TRUEBEAM accelerators). An Electronic Portal Imaging Device was used to perform a set-up verification prior to the delivery of each radiotherapy daily fraction. Fig 1 depicts an example of treatment planning.

## Statistical analysis

Baseline characteristics were described as percentages for categorical data and as mean (SD) for continuous data. Data were compared using Chi-square tests for categorical variables and Student's t-tests for continuous variables. Disease free survival and overall survival were calculated using the Kaplan–Meier method. Log rank test was used to compare survival between 2 groups. All tests were two-sided and P values of less than 0.05 was considered statistically significant. All analyses were performed using IBM SPSS statistics (Version 22.0).

## Follow up

The participants were followed every 3–4 months for the first few years and 6 months for year 4 and 5, and then yearly after that. They were seen by surgeons, medical oncologists and radiation oncologists in separate clinics. At each follow-up, medical history was collected, physical examination was performed, and toxicity was assessed. Mammogram and breast ultrasound were performed 6 months after radiation therapy and then yearly. The late toxicity outcome was assessed by the RTOG/EORTC late radiation morbidity scoring scheme [25].

(A)　　　　　　　　　　　　　　　　　(B)

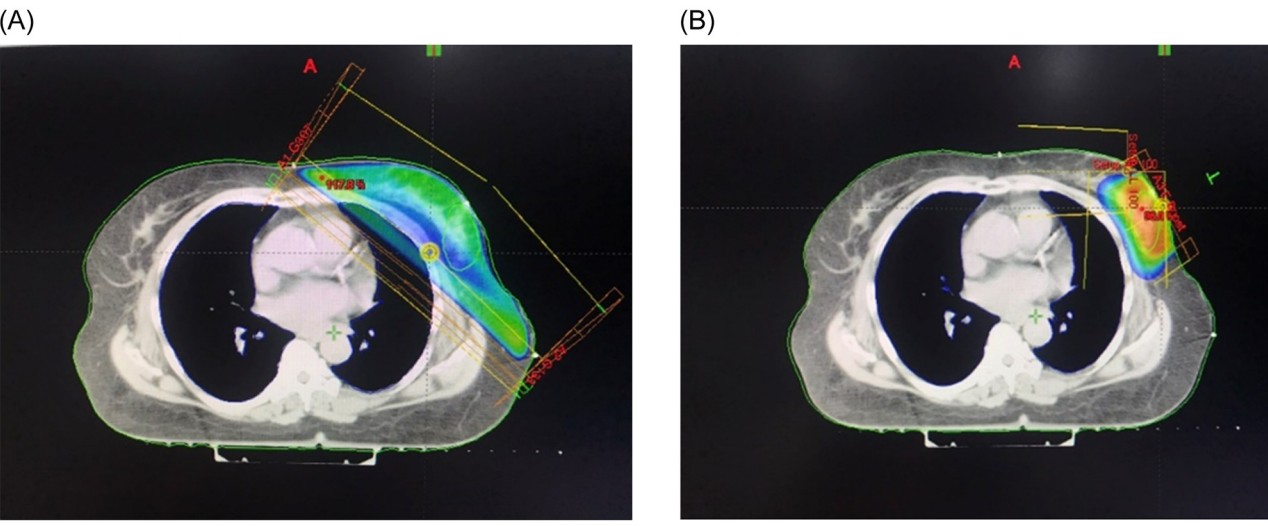

**Fig 1. The treatment planning.** 1A. Whole breast irradiation. 1B. Electron beam boost.

## Results

The median follow-up time was 123 months (range 21–135 months). Totally, 73 participants were enrolled into the study. Thirty-six participants were in Conv-SEQ group and 37 participants were in Hypo-CB group. One participant was lost to follow-up. The baseline characteristics are summarized in Table 1 and are comparable between the 2 study groups. The median age was 52 years (range, 33–76 years) and 47.1% of the participants wereT1 and 84.9% were N0. Eighty nine percent of the participants had Invasive ductal carcinoma and 80.2% of the participants had a negative resected margin (>1 mm width). Seventy-one percent of the participants were ER positive, 24.7% were ER negative, 20.5% were HER2 positive and 64.4% were HER2 negative. Sixty-eight percent of the participants received 2D radiation technique, and 31.5% were treated with 3D radiation technique. Participants who had chemotherapy received 1 of the following regimens: CMF (cyclophosphamide, methotrexate and 5-fluorouracil), AC (doxorubicin and cyclophosphamide) and/or taxane regimen, starting 3–4 weeks after surgery.

## Outcomes

The ipsilateral local recurrence (ILR) was found in 3 participants of which 2 were from the Conv-SEQ group. Two of the participants had invasive cancers, and 1 had a ductal carcinoma in situ. The clinico-pathologic and treatment details of the 3 recurrences are listed in the Table 2. All 3 recurrences were eligible for salvage surgery, either wide excision or mastectomy. At the time of analysis, 82.2% of the participants were alive without disease recurrence. There were 3 participants who had regional and distant metastases and 1 participant developed distant metastases without locoregional recurrence. At 10 years, DFS rate was not significantly different between Hypo-CB and Conv-SEQ (94.4% and 93.9% (p = 0.96), respectively. During this study period, 5 of the participants (6.8%) died. Of these 5 participants, 3 died of breast cancer, and 2 of them were from the hypofractionated group. The other 2 participants died from unknown cause. The 10-year overall survival rate was not significantly different between the 2 groups which had a 10-year overall survival rate of 91.9% for the Hypo-CB group and 91.6% for the Conv-SEQ group (p = 0.792) (Fig 2). Sixty-five (89%) participants had late toxicity. At the time of analysis, there were 5 (7.7%), 56 (86.2%) and 4 (6.2%) participants who had grade

**Table 1. Baseline characteristics by fractionation schedule.**

| Characteristics | No. of participants in the conventional group | No. of participants in the hypofractionated group | p-value |
|---|---|---|---|
| | N = 36 | N = 37 | |
| *Age*: | | | 0.84 |
| 18–40 | 5 (13.9%) | 8 (21.6%) | |
| 41–50 | 10 (27.8%) | 9 (24.3%) | |
| 51–60 | 15 (41.6%) | 15 (40.5%) | |
| > 60 | 6 (16.7%) | 5 (13.6%) | |
| Mean (SD) | 50 (10.4) | 49.8 (9.5) | |
| *Stage*: | | | 0.73 |
| 0 (Tis) | 4 (11.1%) | 3 (8.1%) | |
| I | 13 (36.1%) | 15 (40.5%) | |
| IIA | 19 (52.8%) | 18 (48.7%) | |
| IIB | 0 (0%) | 1 (2.7%) | |
| *Histology*: | | | 0.99 |
| IDC | 31 (86.2%) | 32 (86.5%) | |
| ILC | - | - | |
| DCIS | 4 (11.1%) | 4 (10.8%) | |
| other | 1 (2.7%) | 1 (2.7%) | |
| *Margin*: | | | 0.36 |
| < 1 mm | 8 (22.3%) | 6 (16.2%) | |
| ≥ 1 mm | 28 (77.7%) | 31 (83.8%) | |
| *Tumor grade*: | | | 0.96 |
| Grade I | 6 (16.7%) | 7 (18.9%) | |
| Grade II | 10 (27.8%) | 10 (27.1%) | |
| Grade III | 14 (38.9%) | 16 (43.2%) | |
| Unknown | 6(16.6%) | 4(10.8%) | |
| *Lymphovascular invasion* | | | 0.67 |
| Absent | 12 (33.3%) | 16 (43.2%) | |
| Present | 3 (8.3%) | 3 (8.1%) | |
| Unknown | 21 (58.4%) | 18 (48.7%) | |
| *Ki 67 level* | | | 0.60 |
| <14 | 10 (27.8%) | 14 (37.8%) | |
| ≥15 | 19 (52.8%) | 18 (48.6%) | |
| Unknown | 7 (19.4%) | 5 (13.6%) | |
| *Tumor subtype* | | | 0.48 |
| ER +/ HER2- | 18 (50%) | 20 (54.1%) | |
| ER +/HER2+ | 6 (16.7%) | 5 (13.5%) | |
| ER-/HER2- | 6 (16.7%) | 3 (8.1%) | |
| ER-/HER2+ | 3 (8.3%) | 1 (2.7%) | |
| ER+/HER2 unknowm | 2 (5.6%) | 6 (16.2%) | |
| unknown | 1 (2.7%) | 2 (5.4%) | |
| *Treatment*: | | | 0.22 |
| Surgery | 0 (0%) | 2 (5.4%) | |
| Surgery and chemotherapy | 9 (25%) | 6 (16.2%) | |
| Surgery and hormonal treatment | 14 (38.9%) | 10 (27%) | |
| Surgery, chemotherapy and hormonal treatment | 13 (36.1%) | 19 (51.4%) | |
| *Radiation technique*: | | | 0.57 |
| 2D | 24 (66.6%) | 27 (73%) | |

*(Continued)*

**Table 1.** (Continued)

| Characteristics | No. of participants in the conventional group | No. of participants in the hypofractionated group | p-value |
|---|---|---|---|
|  | N = 36 | N = 37 |  |
| 3D | 12 (33.3%) | 10 (27%) |  |
| *Boost* |  |  | 0.30 |
| Yes | 35 (97.2%) | 37 (100%) |  |
| No | 1 (2.8%) | 0 (0%) |  |

Abbreviations: DCIS, ductal carcinoma in situ; IDC, invasive ductal carcinoma; ILC, invasive lobular carcinoma; ER, Estrogen receptor; HER2, human epidermal growth factor receptor 2; 2D, two-dimensional; 3D, three-dimensional.

0,1,2 skin toxicities, respectively. There was no statistical difference in late toxicity between the 2 groups (p = 0.072).

## Discussion

Multiple randomized phase III studies have confirmed that local control rates for breast cancer increase when boost to the tumor bed in conventional fractionation is used [16–18, 26–29]. A Cochrane systematic review assessed 5 randomized controlled trials of patients with or without boost treatment revealed that local control was significantly better for patients receiving a tumor bed boost compared to patients who did not have a tumor bed boost (HR 0.64, 95% CI 0.55–0.75). However, there were no differences in terms of disease free survival and overall survival between the tumor bed boost and no boost groups (HR 0.94, 95%CI 0.87–1.02 and HR1.04, 95% CI 0.94–1.14, respectively) [30].

Incorporating a CB to the tumor bed in this study increase the radiation dose to the tumor bed and shortened the overall treatment duration. Regarding the effectiveness data of hypofraction with concomitant or simultaneous boost, Formenti et al. [31] enrolled patients to receive 40.5 Gy with 0.5 Gy delivered concomitantly to the tumor bed. There was no local recurrence except 1 patient who had regional recurrence at 1-year of follow-up. Morganti et al. [32] evaluated 2 clinical studies of which 1 of those patients received 44 Gy in 16 fractions to the whole breast and 0.25 Gy concomitant boost. There was no local relapse at median follow-up at 31 months. Corvo et al. [33] used the hypofractionated regimen of 46 Gy in 20 fractions to the whole breast, 4 times a week with a concomitant weekly boost of 1.2 Gy. The authors reported no local recurrent at the median follow up at 33 months. Freedman et al. [20]

**Table 2. Characteristics of the three participants who developed ipsilateral local recurrence.**

| Participant ID number | Stage | Radiation treatment | Systemic treatment | Time to recurrence (month) | Pathology of recurrence |
|---|---|---|---|---|---|
| 34 | pT2N1M0 ER- PR- DISH+ | 60Gy/30Fx | AC x 4 > Tx12 with Trastuzumab | 53 | IDC, ER- PR- Her2 3+ |
| 35 | pT1N0M0 Not available IHC | 52.8Gy/16Fx | CMFx6 | 43 | IDC, ER90% PR90% Her2 2+ |
| 63 | pT1N0M0 ER30% PR- Her2 2+ | 60Gy/30Fx | CMF x 6, Tam | 102 | DCIS, ER95% PR90% |

Abbreviations: DISH: Dual in situ hybridization; AC, doxorubicin and cyclophosphamide; T, paclitaxel; CMF, cyclophosphamide, methotrexate and 5 fluorouracil; Tam, tamoxifen.

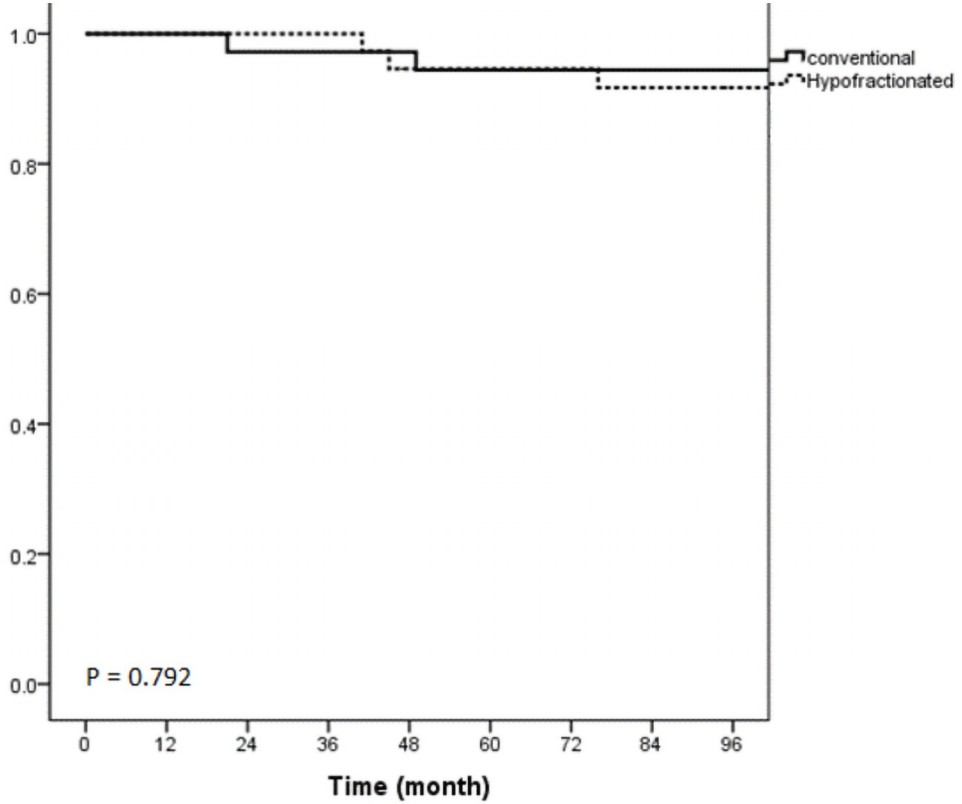

**Fig 2. Kaplan-Meier curve of overall survival rate of both groups.**

reported a 5-year actual rate of local recurrence of 2.7%. In Cante's study [34], 375 patients received 45Gy of the whole breast irradiation with additional 0.25 Gy for concomitant boost. At approximately 5 years of follow-up, there was no local recurrence, and the 5-year DFS and OS were 96.6 and 97.6%, respectively. In Chadha's study [21], 160 early breast cancer patients received 40.5Gy (2.7 Gy per fraction) with a concomitant boost of 0.3 Gy in each fraction. At median follow-up time of 3.5 years, the 5-year OS, DFS and local relapse-free survival were 90%, 97% and 99%, respectively. Other ongoing studies such as RTOG 1005 and HYPOSIB have not yet reported their results [35, 36]. As for our study, the 10-year actual rate of ILR was 4.1% which was quite low. The results were consistent to the aforementioned studies and the 6 pivotal trials.

Several studies have shown that breast cancer cells have lower levels of α/β, and that with an increased BED with SIB technique could improve local tumor control. However, this technique can increase late toxicity which is of concern. Our data revealed that there was no significant late toxicity when the SIB technique was used compared to the conventional scheme.

The IMPORT HIGH trial recently published a 3-year adverse effects of 2,617 breast cancer patients. The authors assessed the dose escalation using SIB or sequential boost after breast conservation surgery. At median follow-up time of 4 year, the rates of moderate and high-grade adverse effects were similar among all treatment groups. There was slight increase in breast induration in the hypofractionated SIB treatment group (53 Gy/15Fx to tumor bed) [37, 38]. The HYPOSIB reported that there was no grade3-4 skin toxicity and the cosmetic outcome was good to excellent at 3 years follow-up [36]. Our results were similar to other studies

as there were no severe toxicity and most of the participants experienced grade 0–1 skin toxicity.

There are some limitations in our study. First, this is not a randomized study and the assignment of the treatment depended on the radiation oncologists. However, the baseline characteristics of both groups were comparable. Second, 69% of our participants received electron tumor bed boost which might result in dose inhomogeneity and non-conformity. Third, we use the surgical scar, the seroma or the distorted tissue plane in case with cavity closure to visualized the CT planning for tumor bed localization. This method was not as precise as the clip-based localization technique.

In terms of the radiation techniques, 60–70% of our participants were treated with conventional technique. Donovan et al. [39] compared between 3D intensity modulated radiotherapy and 2D treatment planning. Although, breast appearance was statistically significantly higher in the 2D treatment arm but there were no significant differences in the patients' quality of life. However, the key strength of the present study was the high rate of study completion (98.6%). As far as we know, this study is the first to evaluate the long-term efficacy of hypofraction with CB. The follow-up period of this study was 10 years post surgery.

In the light of COVID-19 pandemic situation, this radiation protocol may be deemed more appropriate for the patients because there are fewer clinic visits, thus limiting the exposure to COVID-19.

## Conclusion

Our long-term follow up results showed that the incorporation of a tumor bed boost concomitantly with hypofractionated whole breast irradiation is an effective treatment with low local recurrence rate, excellent DFS and OS.

## Acknowledgments

We would like to acknowledge the support of the Division of Therapeutic Radiology and Oncology, Faculty of Medicine, Chulalongkorn University, Bangkok 10330, Thailand.

## Author Contributions

**Conceptualization:** Kitwadee Saksornchai, Prayuth Rojpornpradit.

**Data curation:** Kitwadee Saksornchai, Prayuth Rojpornpradit.

**Formal analysis:** Kitwadee Saksornchai, Prayuth Rojpornpradit.

**Investigation:** Kitwadee Saksornchai, Prayuth Rojpornpradit.

**Methodology:** Kitwadee Saksornchai, Prayuth Rojpornpradit.

**Project administration:** Kitwadee Saksornchai, Prayuth Rojpornpradit.

**Resources:** Kitwadee Saksornchai, Prayuth Rojpornpradit.

**Software:** Kitwadee Saksornchai, Prayuth Rojpornpradit.

**Supervision:** Prayuth Rojpornpradit.

**Validation:** Kitwadee Saksornchai, Prayuth Rojpornpradit.

**Visualization:** Kitwadee Saksornchai, Prayuth Rojpornpradit.

**Writing – original draft:** Kitwadee Saksornchai, Prayuth Rojpornpradit.

**Writing – review & editing:** Kitwadee Saksornchai, Thitiporn Jaruthien, Chonnipa Nanta-
vithya, Kanjana Shotelersuk, Prayuth Rojpornpradit.

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
