## [Decision Letter · Decision Letter 0]

21 Dec 2020

PONE-D-20-35897

Long - term results of hypofractionation with simultaneous integrated boost in early breast cancer: A prospective study.

PLOS ONE

Dear Dr. Saksornchai,

Thank you for submitting your manuscript to PLOS ONE. After careful consideration, we feel that it has merit but does not fully meet PLOS ONE’s publication criteria as it currently stands. Therefore, we invite you to submit a revised version of the manuscript that addresses the points raised during the review process.

The article needs a deep revision. The authors should focus more on the safety, evaluation of the toxicity, potential advantages of the proposed treatment, describing more accurately the impact on patients outcome and the limitations of the study.

We look forward to receiving your revised manuscript.

Kind regards,

Pierpaolo Alongi

Academic Editor

PLOS ONE

Additional Editor Comments:

The article needs a deep revision. The authors should focus more on the safety, evaluation of the toxicity, potential advantages of the proposed treatment, describing more accurately the impact on patients outcome and the limitations of the study.

Journal Requirements:

2. In your Methods section, please provide additional information about the participant recruitment method and the demographic details of your participants. Please ensure you have provided sufficient details to replicate the analyses such as: a) a description of how participants were recruited, and b) descriptions of where participants were recruited and where the research took place.

Reviewers' comments:

Reviewer's Responses to Questions

**Comments to the Author**

1. Is the manuscript technically sound, and do the data support the conclusions?

Reviewer #1: Partly

Reviewer #2: Partly

Reviewer #3: No

2. Has the statistical analysis been performed appropriately and rigorously? 

Reviewer #1: No

Reviewer #2: Yes

Reviewer #3: No

3. Have the authors made all data underlying the findings in their manuscript fully available?

Reviewer #1: No

Reviewer #2: Yes

Reviewer #3: No

4. Is the manuscript presented in an intelligible fashion and written in standard English?

Reviewer #1: Yes

Reviewer #2: Yes

Reviewer #3: No

5. Review Comments to the Author

Reviewer #1: I think that this topic should be related to toxicity and not to efficacy.

However, the Authors could explain the reason to expect an impact with the sib on efficacy: they could talk about the BED in the Materials&Methods and not in the discussion.

Another important topic to underlyne is the boost: i believe that more details about the patient' age sholud be analyzed because of the different impact on efficacy in relation th the age.

Because of one of the benefit of the hypofractionation with SIB is the reduction of overall treatment time, this data must be reported in the results section.

Moreover, data about toxicity are mandatory, as shown in the introduction section (line 54-58): could you describe the tolerance of your patients?

M&M

-The Authors stated that patients are T1-3 N0-1 (line 73), but this doesn't reflect what they said at line 79.

-How did you calculate the follow up? Since surgery?the end of RT? Please explain

-Could You describe the follow up of your patients? Did the Surgeon or The Radiation Oncologist follow the patients?

-How did you make the clinical diagnosis of the recurrence?

-What kind of baseline characteristics did you collect?(i.e.: comorbidities, tumor biology, ki 67...ecc...)

-The total number of the patients is low: why did the patients decline to take part in teh study? You said that both radiotherapy fractionation schemes are standard of care...

RESULTS

-line 120: please 68.5 % should be reported in letters because at the beginning of the sentence.

-The baseline characteristics should be expanded with tumor-characteristics (biology, KI 67, lymphovascular invasion...): this could justify (or not) the low number of recurrence

-A stratification by age sould as follows should be reported: patients aged ≤40 years, 41–50 years, 51–60 years, >60 years.

-In the section of "outcomes" you have to report both with absolute number and % (not only one). Moreover the recurrence must be describe in details: i don't understand the value of 17.8% (what kind of recurrence?global?). If more details are shown with the absolute number, it could be easier to understand the results.

-In table 2, patient n 35 received 54.4Gy/16 fr, but this fractionation doesn't correspond to your hypofractionation of 43.2 in 16 fraction with a sib of 0.6 Gy for each fraction: please explain.

-I understand that the events number are low, but you could report the Kaplan-Meier curves because your follow-up is very long.

DISCUSSION

This section should explain the impact of the SIB on efficacy-ouitcomes: you should expand what you've reported at lines 160-163.

please, correct line 17 (remove "In" athe the beginning).

Finally, at line 181 you introduced toxicity, but this is not a topic of your study: i believe that this is the most importanty aspect to explore in the setting of hypofractionation with SIB. Please expand.

Reviewer #2: This is a single-institution prospective non-randomized trial, with 10 years follow-up, of 73 early-stage breast cancer patients treated with breast-conserving sugery followed by whole-breast irradiation and a boost to the tumour bed, using two-dimensional (2D) or three-dimensional (3D) conformal radiation technique; and adjuvant systemic treatment.

36 patients received conventional radiation to the whole breast (50 Gy in 25 fractions) with a sequential boost to the tumour bed (10–16 Gy) (Conv-SEQ), while the other 37 patients received a hypofractionated scheme (43.2 Gy in 16 fractions) with an additional daily simultaneous integrated boost (SIB) of 0.6 Gy (Hypo-SIB).

The long-term follow-up data demonstrate no significant differences in the 10-year disease free survival (DFS) (93.9% vs 94.4%, p = 0.96) or 10-year overall survival rates (OS) (91.6% vs 91.9%, p = 0.792) between the conventional or hypofractionated groups.

The following modifications will improve upon your paper:

Major criticisms:

1. Materials and methods, Study population: the Authors report that they enrolled patients treated from 2005 to 2009 in the study, while in the abstract they refer to the period between October 2009 and June 2010. Please correct it.

2. Materials and methods, Study population: “The eligibility criteria included patients aged 18 or older, with resected tumour margins wider than 1 mm, with no lymph node involvement”. But on page 4, line 73, the Authors described to enrolled “all patients with operable invasive breast cancer (T1-3N0-1M0)...”.

I suggest to better define the eligible criteria for patients included in this analysis.

3. Materials and methods: how was the follow-up performed? (surveillance time, radiological examinations, etc.)

4. On pag 4, line 77, the Authors state that “assignment of patients to each arm was made at the discretion of the treating radiation oncologist”. This statement is inaccurate and I suggest to better define the assignment criteria in the two treatment schedules. This is the weakest point of this study - SELECTION BIAS.

5. In the results section the reported baseline characteristics are not clear (for example from lines 119 to 120 the Authors reported that “71.2% of the patients were ER positive and 24.7% were ER negative”, but the sum of the percentages is not 100%).

I suggest to add a table with patient characteristics and their correct percentages.

6. Results: the Authors described the different schedules of adjuvant chemotherapy. But, from line 121 to 124, they wrote “All patients received one of the following adjuvant chemotherapy...”, while in table 1 we can see that not all patients were treated with chemotherapy. Please correct this information.

7. The study focused on survival outcomes. Taking into account the long follow-up, it would be interesting to report data regarding treatment-related toxicity or cosmesis. If available, I suggest to add it.

Minor limitations

1. Table 1: reported number of patients and percentages approximation are wrong. For examples in stage and treatment sections the number of patients underwent conventional or hypofractionated treatment are 39 and 34, respectively.

2. I suggest to add the pathological and biological characteristics in table 1.

3. In the outcomes section (from line 141 to 142): reported rates of recurrence and metastasis are not clear.

4. Line 146: reported percentages refer to 10-year OS for Conv-SEQ is different from that described in the abstract. Please correct it.

5. Table 2: all abbreviations and acronyms should be explained at first use. DISH was not previously defined.

Reviewer #3: In the current Long - term results of hypofractionation with simultaneous integrated boost in early breast cancer the Authors report the findings of a prospective study comparing conventional versus hypofractionation. There are several limitations such as the limited cohort of patients to drawn any conclusion.

I believe that the results of the present analysis are not so useful in the daily clinical practice (there are several randomized trials that showed not inferiority of Hypo versus conventional)

6. PLOS authors have the option to publish the peer review history of their article (what does this mean?). If published, this will include your full peer review and any attached files.

Reviewer #1: No

Reviewer #2: No

Reviewer #3: No

---

## [Author Response · Author response to Decision Letter 0]

22 Apr 2021

Reviewer #1

I think that this topic should be related to toxicity and not to efficacy.

However, the Authors could explain the reason to expect an impact with the sib on efficacy: they could talk about the BED in the Materials&Methods and not in the discussion.

Another important topic to underline is the boost: i believe that more

details about the patient' age should be analyzed because of the different impact on efficacy in relation the the age. Because of one of the benefit of the hypofractionation with SIB is the reduction of overall treatment time, this data must be reported in the results section.

Moreover, data about toxicity are mandatory, as shown in the

introduction section (line 54-58): could you describe the tolerance of

your patients?

Response:

I have added the BED in the Materials and Methods section.

I have added the late toxicity information.

I have added the age and boost information in the patient characteristic.

The Authors stated that patients are T1-3 N0-1 (line 73), but this doesn't reflect what they said at line 79.

Response: 

I have changed the sentence of patient characteristic that included in my study.

How did you calculate the follow up? Since surgery? The end of RT?

Please explain

Response: 

I calculated follow up since surgery.

Could You describe the follow up of your patients? Did the Surgeon or

The Radiation Oncologist follow the patients?

Response: 

I have added the follow up section which explain that the patients were followed up by surgeons, radiation oncologists and medical oncologists.

How did you make the clinical diagnosis of the recurrence?

Response: 

We did the imaging (CT scans) and the tissue biopsy.

What kind of baseline characteristics did you collect?(i.e.:comorbidities, tumor biology, ki 67…ecc...)

Response: 

I have added the patient characteristic in table 1.

The total number of the patients is low: why did the patients decline to take part in the study? You said that both radiotherapy fractionation schemes are standard of care…

Response: 

We calculated the number of patients by using match pair method, with a 90% of power. Pearson Chi-square test was used for testing the difference between groups with an alpha value of 0.05.

RESULTS

line 120: please 68.5 % should be reported in letters because at the beginning of the sentence.

Response: 

I have changed it to be the letters.

The baseline characteristics should be expanded with tumor characteristics

(biology, KI 67, lymphovascular invasion...): this could justify (or not) the low number of recurrence

Response: I have added the information of baseline characteristics.

-A stratification by age sould as follows should be reported: patients aged ≤40 years, 41–50 years, 51–60 years, >60 years.

Response: 

I have done it in the table 1.

In the section of "outcomes" you have to report both with absolute number and % (not only one). Moreover the recurrence must be describe in details: i don't understand the value of 17.8% (what kind of recurrence? global?). If more details are shown with the absolute number, it could be easier to understand the results.

Response: 

I have added the absolute outcome.

In table 2, patient n 35 received 54.4Gy/16 fr, but this fractionation doesn't correspond to your hypofractionation of 43.2 in 16 fraction with a sib of 0.6 Gy for each fraction: please explain.

Response: 

I am sorry it is wrong number.

I understand that the events number are low, but you could report the Kaplan-Meier curves because your follow-up is very long.

Response: 

I have added the graph in the paper.

DISCUSSION

This section should explain the impact of the SIB on efficacy outcomes:

you should expand what you've reported at lines 160-163. please, correct line 17 (remove "In" at the the beginning).

Finally, at line 181 you introduced toxicity, but this is not a topic of your

study: i believe that this is the most importanty aspect to explore in the

setting of hypofractionation with SIB. Please expand.

Response: 

I have added the toxicity information.

Reviewer #2

Major criticisms:

1. Materials and methods, Study population: the Authors report that they

enrolled patients treated from 2005 to 2009 in the study, while in the

abstract they refer to the period between October 2009 and June 2010.

Please correct it.

Response: I have corrected the period time to be from 2009 to 2010

2. Materials and methods, Study population: “The eligibility criteria

included patients aged 18 or older, with resected tumour margins wider

than 1 mm, with no lymph node involvement”. But on page 4, line 73,

the Authors described to enrolled “all patients with operable invasive

breast cancer (T1-3N0-1M0)...”.

I suggest to better define the eligible criteria for patients included in this

analysis.

Response: I have corrected the eligibility of patients to be

Eligibility criteria included patients with operable invasive breast cancer

(T1-3N0-1M0), aged 18 or older with resected tumor margins wider than

1 mm.

3. Materials and methods: how was the follow-up performed?

(surveillance time, radiological examinations, etc.)

Response: I have added the follow up examinations

Follow up examinations

Patients were followed up every 3-4 months for the first few years and 6 months for year 4 and 5 and then yearly. They were seen by surgeons, radiation oncologist and medical oncologist in the separate clinic. At each follow-up, medical history, physical examination and toxicity were performed and assessed. Mammogram and breast ultrasound was performed 6 months after radiation therapy and then yearly.

4. On page 4, line 77, the Authors state that “assignment of patients to

each arm was made at the discretion of the treating radiation oncologist”. This statement is inaccurate and I suggest to better define the assignment criteria in the two treatment schedules. This is the weakest point of this study - SELECTION BIAS.

Response: I would add this as the limitation of the study. But we found that the baseline characteristics of both groups are comparable of the two groups. I have changed the statement to be the treatment schedules were assigned by the radiation oncologists.

5. In the results section the reported baseline characteristics are not clear (for example from lines 119 to 120 the Authors reported that “71.2% of the patients were ER positive and 24.7% were ER negative”, but the sum of the percentages is not 100%). I suggest to add a table with patient

characteristics and their correct percentages.

Response: I added patient characteristics in the table 1

For the ER status of the patients: 71.2 % was ER positive and 24.7% was ER negative. The rest of the patients were unknown.

6 Results: the Authors described the different schedules of adjuvant chemotherapy. But, from line 121 to 124, they wrote “All patients received one of the following adjuvant chemotherapy...”, while in table 1 we can see that not all patients were treated with chemotherapy. Please correct this information.

Response: I have corrected this sentence

For those patients who received adjuvant chemotherapy, they receive

one of the following regimens

7. The study focused on survival outcomes. Taking into account the long follow-up, it would be interesting to report data regarding treatment related toxicity. If available, I suggest to add it.

Response: I have added the information of late toxicity outcome.

Minor limitations

1. Table 1: reported number of patients and percentages approximation are wrong. For examples in stage and treatment sections the number of patients underwent conventional or hypofractionated treatment are 39 and 34, respectively.

Response: I have added the table of patient characteristic.

2. I suggest to add the pathological and biological characteristics in table

1.

Response: I have added those information in table 1

3. In the outcomes section (from line 141 to 142): reported rates of

recurrence and metastasis are not clear.

Response: I have added and clarified the information.

4. Line 146: reported percentages refer to 10-year OS for Conv-SEQ is

different from that described in the abstract. Please correct it.

Response: I have corrected the number in the abstract to be 91.6% vs 91.9%.

5. Table 2: all abbreviations and acronyms should be explained at first use. DISH was not previously defined.

Response: I have explained all abbreviations

---

## [Decision Letter · Decision Letter 1]

13 Jul 2021

PONE-D-20-35897R1

Long - term results of hypofractionation with simultaneous integrated boost in early breast cancer: A prospective study.

PLOS ONE

Dear Dr. Saksornchai,

Thank you for submitting your manuscript to PLOS ONE. After careful consideration, we feel that it has merit but does not fully meet PLOS ONE’s publication criteria as it currently stands. Therefore, we invite you to submit a revised version of the manuscript that addresses the points raised during the review process.

ACADEMIC EDITOR: The article still presents several critical points to review. Please find the reviewer's suggestions and give a substantial response about the limitations of the study and modifications requested. 

We look forward to receiving your revised manuscript.

Kind regards,

Pierpaolo Alongi

Academic Editor

PLOS ONE

Journal Requirements:

Additional Editor Comments (if provided):

Reviewers' comments:

Reviewer's Responses to Questions

**Comments to the Author**

1. If the authors have adequately addressed your comments raised in a previous round of review and you feel that this manuscript is now acceptable for publication, you may indicate that here to bypass the “Comments to the Author” section, enter your conflict of interest statement in the “Confidential to Editor” section, and submit your "Accept" recommendation.

Reviewer #4: (No Response)

Reviewer #5: All comments have been addressed

2. Is the manuscript technically sound, and do the data support the conclusions?

Reviewer #4: Yes

Reviewer #5: Partly

3. Has the statistical analysis been performed appropriately and rigorously? 

Reviewer #4: (No Response)

Reviewer #5: Yes

4. Have the authors made all data underlying the findings in their manuscript fully available?

Reviewer #4: (No Response)

Reviewer #5: Yes

5. Is the manuscript presented in an intelligible fashion and written in standard English?

Reviewer #4: (No Response)

Reviewer #5: No

6. Review Comments to the Author

Reviewer #4: Although it has some weaknesses for the number of patients, treatment assignment criteria, and dosimetric gap, it is still a prospective study on a current topic of interest to PLOS ONE readers.

The authors responded well to the first reviews. However, I suggest other revisions to be performed before a final decision on the paper.

INTRODUCTION

1) Line 49. The statement “The conventional radiation to the whole breast is 45–50 Gy, administered in 1.8–2 Gy per fraction doses over 5–6 weeks” needs bibliographic reference.

2) Line 53. I suggest to add reference FAST Trial doi: 10.1200/JCO.19.02750. Epub 2020 Jul 14.

3) I recommend adding the data from Krug's recent article. “Adjuvant hypofractionated radiotherapy with simultaneous integrated boost after breast-conserving surgery: results of a prospective trial” https://doi.org/10.1007/s00066-020-01689-7

MATERIALS

1) Line 99. The paragraph “Radiotherapy” is lacking.

I suggest to add the data of the LinAc used.

2) Line 103. “The details of the radiation technique have been described previously. [(21]) “

The reference provided describes a concomitant boost, not a simultaneous boost.

I suggest to better describe the technique used and specify whether it is a concomitant or simultaneous boost.

Although many authors use the 2 terms indistinctly, they are different.

Concomitant boost treatment involves performing 2 different plans in the same fraction.

Simultaneous boost treatment involves performing a single plane through intensity-modulated techniques.

3) I suggest to add contouring information and treatment plan images.

RESULTS

1)Collecting dosimetric data and correlating them with recurrence or skin toxicity could consolidate the study.

DISCUSSION

1) Line 190. Several studies described use concomitant boost, but you refer to simultaneous boost. I recommend changing the terms.

2) I suggest extending the discussion by analysing the advantages and disadvantages of 2D and 3D versus intensity-modulated techniques. I would recommend a few references on this topic.

Donovan E, Bleakley N, Denholm E, Evans P, Gothard L, Hanson J, et al. Randomised trial of standard 2D radiotherapy (RT) versus intensity modulated radiotherapy (IMRT) in patients prescribed breast radiotherapy. Radiother Oncol J Eur Soc Ther Radiol Oncol 2007;82:254–64.

https://doi.org/10.1016/j.radonc.2006.12.008.

van der Laan HP, Dolsma WV, Schilstra C, Korevaar EW, de Bock GH, Maduro JH, et al. Limited benefit of inversely optimised intensity modulation in breast conserving radiotherapy with simultaneously integrated boost. Radiother Oncol J Eur Soc Ther Radiol Oncol 2010;94:307– 12. https://doi.org/10.1016/j.radonc.2010.01.024.

Joseph B, Farooq N, Kumar S, Vijay CR, Puthur KJ, Ramesh C, et al. Breast-conserving radiotherapy with simultaneous integrated boost; field-in-field three-dimensional conformal radiotherapy versus inverse intensity-modulated radiotherapy - A dosimetric comparison: Do we need intensity-modulated radiotherapy? South Asian J Cancer 2018;7:163–6. https://doi.org/10.4103/sajc.sajc_82_18.

Cozzi L, Lohr F, Fogliata A, Franceschini D, De Rose F, Filippi AR, et al. Critical appraisal of the role of volumetric modulated arc therapy in the radiation therapy management of breast cancer. Radiat Oncol Lond Engl 2017;12. https://doi.org/10.1186/s13014-017-0935-4.

Lee H-H, Hou M-F, Chuang H-Y, Huang M-Y, Tsuei L-P, Chen F-M, et al. Intensity modulated radiotherapy with simultaneous integrated boost vs. conventional radiotherapy with sequential boost for breast cancer - A preliminary result. Breast Edinb Scotl 2015;24:656–60. https://doi.org/10.1016/j.breast.2015.08.002.

3) I recommend emphasising the importance of this study in the perspective of the Covid-19 pandemic period.

CONCLUSION

1) Also in the conclusion and thus in the title, you should better define whether it is a concurrent or simultaneous boost.

Reviewer #5: Dear Authors,

the topic is interesting due to a lack from randomized studies.Unfortunately, even if your effort is remarkable, the sample size of the population, the retrospective nature of your study and the absence of statistical approach to increase the homogeneity of the population (pooled analysis or similar) are strong limitations. Furthermore some part of the paper are not written in a fluent English and somewhere grammatical errors are present and this lowered the quality of the paper.

7. PLOS authors have the option to publish the peer review history of their article (what does this mean?). If published, this will include your full peer review and any attached files.

Reviewer #4: **Yes: **Antonio Piras

Reviewer #5: No

---

## [Author Response · Author response to Decision Letter 1]

6 Sep 2021

Reviewer #4 

INTRODUCTION

1) Line 49. The statement “The conventional radiation to the whole breast is 45–50 Gy, administered in 1.8–2 Gy per fraction doses over 5–6 weeks” needs bibliographic reference.

Response: I have added the references which are 

-Fisher B, Anderson S, Bryant J, et al. Twenty-year follow-up of a randomized trial comparing total mastectomy, lumpectomy, and lumpectomy plus irradiation for the treatment of invasive breast can- cer. N Engl J Med 2002;347:1233-41. 

- Veronesi U, Luini A, Del Vecchio M, et al. Radiotherapy after breast-preserving surgery in women with localized cancer of the breast. N Engl J Med 1993;328:1587- 91. 

2) Line 53. I suggest to add reference FAST Trial doi: 10.1200/JCO.19.02750. Epub 2020 Jul 14.

Response: I have added FAST trial reference.

3) I recommend adding the data from Krug's recent article. “Adjuvant hypofractionated radiotherapy with simultaneous integrated boost after breast-conserving surgery: results of a prospective trial” https://doi.org/10.1007/s00066-020-01689-7

Response: Thank you, I have added the data from Krug’ article as you suggested.

MATERIALS

1) Line 99. The paragraph “Radiotherapy” is lacking. I suggest to add the data of the Linac used.

Response: I have added the data regarding our treatment machines.

2) Line 103. “The details of the radiation technique have been described previously. [(21])” The reference provided describes a concomitant boost, not a simultaneous boost. I suggest to better describe the technique used and specify whether it is a concomitant or simultaneous boost. Although many authors use the 2 terms indistinctly, they are different. Concomitant boost treatment involves performing 2 different plans in the same fraction. Simultaneous boost treatment involves performing a single plane through intensity-modulated techniques.

Response: Thank you for the comment. Our study, we use concomitant boost and I have put more detailed about the technique in this paper. I also added the treatment planning image.

3) I suggest to add contouring information and treatment plan images.

Response: I have added the image. (The 3D planning) There are whole breast and electron boost planning.

RESULTS

1) Collecting dosimetric data and correlating them with recurrence or skin toxicity could consolidate the study.

Response: We really appreciate your suggestion. Unfortunately, 60% of our patients were treated with 2D techniques. As a result, the collection of dosimetric data may not be complete. However, we made every effort to make it as prescribed in the radiation technique. For whole breast irradiation, photons 6 MV with a source-to-axis distance of 100 cm were used in all patients. Dose homogeneity of breast was achieved by using wedges. For tumor bed boost, an en-face electron beam with appropriate energy was used, prescribing at the 90% isodose line. 

DISCUSSION

1) Line 190. Several studies described use concomitant boost, but you refer to simultaneous boost. I recommend changing the terms.

Response: I have changed it to be a concomitant boost.

2) I suggest extending the discussion by analysing the advantages and disadvantages of 2D and 3D versus intensity-modulated techniques. I would recommend a few references on this topic.

Response: Thank you very much for all the references. I have added those related information in the manuscript.

3) I recommend emphasising the importance of this study in the perspective of the Covid-19 pandemic period.

Response: I have added the sentence that this paper could be used in the context of the Covid-19 situation.

CONCLUSION

1) Also in the conclusion and thus in the title, you should better define whether it is a concurrent or simultaneous boost.

Response: I have changed into the concomitant boost.

Reviewer 5#

the topic is interesting due to a lack from randomized studies. Unfortunately, even if your effort is remarkable, the sample size of the population, the retrospective nature of your study and the absence of statistical approach to increase the homogeneity of the population (pooled analysis or similar) are strong limitations. Furthermore some part of the paper are not written in a fluent English and somewhere grammatical errors are present and this lowered the quality of the paper.

Response: We are grateful for your comment. Also there is the limitation of the study, we would like to report our long-term clinical and toxicity data which is the key strength of our study. This study will support the use of hypofractionation with concomitant boost in breast cancer. We also submit our manuscript for English editing in order to improve the quality of the paper.

---

## [Editor Report · Decision Letter 2]

22 Sep 2021

Long-term results of hypofractionation with concomitant boost in patients with early breast cancer: A prospective study.

PONE-D-20-35897R2

Dear Dr. Saksornchai,

We’re pleased to inform you that your manuscript has been judged scientifically suitable for publication and will be formally accepted for publication once it meets all outstanding technical requirements.

Kind regards,

Pierpaolo Alongi

Academic Editor

PLOS ONE
---

## [Editor Report · Acceptance letter]

24 Sep 2021

PONE-D-20-35897R2 

Long-term results of hypofractionation with concomitant boost in patients with early breast cancer: A prospective study. 

Dear Dr. Saksornchai:

I'm pleased to inform you that your manuscript has been deemed suitable for publication in PLOS ONE. Congratulations! Your manuscript is now with our production department. 

Kind regards, 

on behalf of

Dr. Pierpaolo Alongi 

Academic Editor

PLOS ONE